# Exploring the Potential Role of Upper Abdominal Peritonectomy in Advanced Ovarian Cancer Cytoreductive Surgery Using Explainable Artificial Intelligence

**DOI:** 10.3390/cancers15225386

**Published:** 2023-11-13

**Authors:** Alexandros Laios, Evangelos Kalampokis, Marios Evangelos Mamalis, Amudha Thangavelu, Richard Hutson, Tim Broadhead, David Nugent, Diederick De Jong

**Affiliations:** 1Department of Gynaecologic Oncology, St James’s University Hospital, Leeds LS9 7TF, UK; amudhathangavelu@nhs.net (A.T.); richard.hutson@nhs.net (R.H.); tim.broadhead@nhs.net (T.B.); david.nugent@nhs.net (D.N.); diederick.dejong@nhs.net (D.D.J.); 2Information Systems Lab, Department of Business Administration, University of Macedonia, 54636 Thessaloniki, Greece; ekal@uom.edu.gr (E.K.); bad22019@uom.edu.gr (M.E.M.); 3Center for Research & Technology HELLAS (CERTH), 6th km Charilaou-Thermi Rd, 57001 Thessaloniki, Greece

**Keywords:** epithelial ovarian cancer, complete cytoreduction, upper abdominal peritonectomy, machine learning, explainable artificial intelligence, survival

## Abstract

**Simple Summary:**

The Surgical Complexity Score (SCS) has been widely used to reflect the surgical effort during advanced stage epithelial ovarian cancer (EOC) cytoreduction. However, not all surgical procedures are described by this score. Using artificial intelligence, we developed and explained an algorithm that weighted the importance of all surgical procedures for the prediction of complete cytoreduction (CC0). We identified upper abdominal peritonectomy (UAP) as the most salient procedural predictor of CC0, followed by pelvic and para-aortic lymph node dissection and ileocecal resection/right hemicolectomy. The UAP was predictive of poorer progression-free survival but not overall survival. The SCS did not impact survival. We advocate thorough early inspection of the upper abdominal quadrants to ensure that CC0 is achievable.

**Abstract:**

The Surgical Complexity Score (SCS) has been widely used to describe the surgical effort during advanced stage epithelial ovarian cancer (EOC) cytoreduction. Referring to a variety of multi-visceral resections, it best combines the numbers with the complexity of the sub-procedures. Nevertheless, not all potential surgical procedures are described by this score. Lately, the European Society for Gynaecological Oncology (ESGO) has established standard outcome quality indicators pertinent to achieving complete cytoreduction (CC0). There is a need to define what weight all these surgical sub-procedures comprising CC0 would be given. Prospectively collected data from 560 surgically cytoreduced advanced stage EOC patients were analysed at a UK tertiary referral centre.We adapted the structured ESGO ovarian cancer report template. We employed the eXtreme Gradient Boosting (XGBoost) algorithm to model a long list of surgical sub-procedures. We applied the Shapley Additive explanations (SHAP) framework to provide global (cohort) explainability. We used Cox regression for survival analysis and constructed Kaplan-Meier curves. The XGBoost model predicted CC0 with an acceptable accuracy (area under curve [AUC] = 0.70; 95% confidence interval [CI] = 0.63–0.76). Visual quantification of the feature importance for the prediction of CC0 identified upper abdominal peritonectomy (UAP) as the most important feature, followed by regional lymphadenectomies. The UAP best correlated with bladder peritonectomy and diaphragmatic stripping (Pearson’s correlations > 0.5). Clear inflection points were shown by pelvic and para-aortic lymph node dissection and ileocecal resection/right hemicolectomy, which increased the probability for CC0. When UAP was solely added to a composite model comprising of engineered features, it substantially enhanced its predictive value (AUC = 0.80, CI = 0.75–0.84). The UAP was predictive of poorer progression-free survival (HR = 1.76, CI 1.14–2.70, P: 0.01) but not overall survival (HR = 1.06, CI 0.56–1.99, P: 0.86). The SCS did not have significant survival impact. Machine Learning allows for operational feature selection by weighting the relative importance of those surgical sub-procedures that appear to be more predictive of CC0. Our study identifies UAP as the most important procedural predictor of CC0 in surgically cytoreduced advanced-stage EOC women. The classification model presented here can potentially be trained with a larger number of samples to generate a robust digital surgical reference in high output tertiary centres. The upper abdominal quadrants should be thoroughly inspected to ensure that CC0 is achievable.

## 1. Introduction

In the western world, epithelial ovarian cancer (EOC) is the fifth most common cause of women’s cancer-related death [1]. Most women are diagnosed at an advanced stage mainly due to the lack of sufficient diagnostic tools (stage III or IV). The current gold standard treatment is cytoreductive surgery combined with carboplatin and paclitaxel chemotherapy and subsequent maintenance therapy [2,3]. Such complex treatment algorithms often require extensive surgical procedures including peritoneal stripping, diaphragmatic, splenic, liver, and gastrointestinal resections [4,5]. Complete cytoreduction (CC0) and chemotherapy response appear to be the most critical prognostic factors [6].

Achieving CC0 frequently requires targeted maximal effort. Previous attempts to describe the extent of cytoreductive surgery led to the development of the surgical complexity score (SCS), which best combined the numbers with the complexity of the procedures [6]. Nevertheless, not all potential surgical procedures are described by this score. Lately, the European Society for Gynecologic Oncology (ESGO) has established ten quality indicators (QIs), based on the standards of practice to audit and improve advanced EOC surgery [7]. Three of these QIs were outcome indicators related to achievement of CC0. In the complex environment of the operating room, CC0 is not always realized. Inconsistency among surgeons in the interpretation of the size of residual disease has been reported, prompting accurate documentation of operative findings and outcomes in the surgical notes [8]. The QI8, a process indicator was related to prospective recorded information from an exhaustive list of structured surgical procedures as “minimum required elements in operative reports” [9]. There is a need to define what weight all these surgical procedures comprising CC0 would be given. Therefore, most surgeons should regularly seek objective but personalised strategies to evaluate their cytoreductive outcomes.

In the era of precision oncology, Artificial Intelligence (AI) could potentially support clinicians in making meaningful predictions of the surgical outcomes for quality improvement and delivery of modern ovarian cancer care [10]. We previously employed such innovative solutions to predict outcomes of cytoreductive surgery in advanced EOC [11,12]. Herein, we developed an AI algorithm to support the weighted importance of all surgical procedures performed at EOC cytoreductive surgery for CC0 forecasting. Using eXplainable Artificial Intelligence (XAI), we examined and interpreted the most salient procedural interactions to explain the overall model predictive performance.

## 2. Materials and Methods

The study was a single-center retrospective cohort study including patients treated at our ESGO accredited center of excellence for advance ovarian cancer surgery between 2014–2019. All consecutive incoming women with newly diagnosed advanced stage EOC who underwent surgery during their primary therapy were included in the study. Exclusion criteria included women < 18 years at first diagnosis, women with relapsed EOC or receiving palliative surgery, women with non-epithelial tumours, and those presenting at first diagnosis with early stage EOC. The patient cohort, the MDT consensus and the hospital setting have been previously described in detail [12,13]. All operations were carried out via a midline laparotomy by a team of gynaecological and, when necessary, hepatobiliary, or colorectal surgeons with an attempt to achieve total macroscopic clearance. Early intra-operative assessment of tumour dissemination was routinely performed and retrospectively documented in the operative notes prior to textual data entry in the ovarian cancer database. Ethics board approval was obtained through the Leeds Teaching Hospitals Trust (MO20/133163/18.06.20). The study was added to the UMIN/CTR Trial Registry (UMIN000049480).

The operative report was a frank adaptation of the structured ESGO ovarian cancer operative report template that included an exhaustive list of pelvic, lower abdomen and upper abdomen surgical procedures [8]. All the regions of the abdominal and pelvic cavity (ovaries, tubes, uterus, pelvic peritoneum, paracolic gutters, anterior parietal peritoneum, mesentery, peritoneal surface of the colon and bowel, liver, spleen, greater and lesser omentum, hepatic port hepatic, stomach, Morrison’s pouch, lesser sac, surface of both hemi diaphragms, pelvic and para-aortic lymph nodes, and if applicable pleural cavity) was evaluated and described [13]. During the study years, systematic pelvic and para-aortic lymph node dissection or sampling was routinely performed, particularly in the presence of bulky lymph nodes. When applicable, the size and location of residual disease at the end of the operation, and the reasons for not achieving complete cytoreduction were reported. An ESGO-approved template was available on the ESGO website (https://guidelines.esgo.org/, accessed on 23 April 2023).

Two separate analyses were performed. Firstly, all cases were analysed to audit the trends of surgical procedures performed overtime in both the primary and interval debulking setting. Secondly, the most important predictive feature was interrogated against commonly used engineered features including the peritoneal carcinomatosis index (PCI) and the intra-operative mapping for ovarian cancer (IMO) score, in addition to the SCS. The PCI and IMO scores were calculated at the beginning of surgery to describe the intra-operative location of the disease [14,15]. We did not perform a propensity score matching, as recent evidence suggests the performance of these procedures does not significantly change in the interval cytoreductive surgery group [16].

Descriptive statistics were used to summarize the clinical characteristics of patients and their respectful cytoreductions. Continuous variables were summarized with means, standard deviations, medians, and ranges. The Kruskall-Wallis test was used to compare groups with respect to median values. Categorical variables were summarized with counts and percent. The Fisher’s exact test was used to compare groups with respect to categorical variables. Progression-free survival (PFS) was defined as the time (months) from the date of initial diagnosis to the date of progression or recurrence. Patients who were alive without progression or recurrence were censored on the date of last clinical assessment. Overall survival (OS) was defined as the time (months) from the date of initial diagnosis to the date of death. Patients who were alive were censored on the date of last follow-up. We used the Kaplan and Meier (K-M) method to estimate median PFS and OS stratified by various potential prognostic factors and the log-rank test to detect associations between variables and outcomes. Multivariate analysis using the Cox proportional hazards method was performed to identify potential independent risk factors for recurrence and mortality. Pearson’s correlation ($r^{2}$) was used to describe the associations amongst numerical variables and heatmaps were produced to illustrate the correlations. All tests were two-sided, and significance was determined at the 0.05 level.

### Model Development

The eXtreme Gradient Boosting (XGBoost) algorithm was employed to model the features [17]. This combines all the generated hypotheses of weak learning algorithms into a single hypothesis to boost performance. The combined effect of eight parameters to maximize model performance was investigated by evaluating a grid of combinations of values using Scikit-learn’s GridSearchCV function.

The dataset was split into training and test cohorts (70%:30% ratio). A five-fold stratified cross-validation (CV) was performed and stratified folds were constructed to overcome data imbalance. The CV was iterated to decrease both variance and bias. Model performance was assessed by measuring the total area under the receiver-operating curve (AUC). Receiver operating characteristic (ROC) and Precision-Recall curves and state-of-art scores were used for performance metrics.

To explain the predictive model, the artificial intelligence SHapley Additive exPlanations (SHAP) framework was employed. The methodology enhances interpretability by computing the importance values for each feature on individual predictions; in other words, it explains how much the presence of a feature contributes to the model’s overall prediction [18]. The framework interprets the model of the entire cohort as a linear function of features. In this way, it explains how much the presence of a feature contributes to the model’s overall predictions. Visual quantification of the model prediction was demonstrated by producing (a) SHAP summary plots for the global (cohort) explanation of the results; (b) SHAP dependence plots of the critical risk features pertinent to the prediction. The Python language Programming Software available at http://www.python.org, accessed on 12 July 2023 was used for the analyses.

## 3. Results

The study enrolled 560 EOC patients. The patient-specific descriptive statistics have been recently published ([12] and Appendix A Table A1). The descriptive of the performed surgical sub-procedures is shown on Table 1. The patients were followed-up until April 2022. Several upper abdominal procedures including wedge liver resection, diaphragmatic stripping, splenectomy, UAP, cholecystectomy, stomach resection was statistically significant between the CC0 and not CC0 groups.

The model performance for the above threshold prediction was moderate-to-high (AUC 0.63, 95% CI 0.60–0.67; AP 0.44, 95% CI 0.41–0.48) (Figure 1). To promote reproducibility, the optimal set of model parameters were the following: XGBoost: {“colsample_bylevel”: 1, “gamma”: 0.7, “learning_rate”: 0.01, “max_delta_step”: 1, “max_depth”: 5, “min_child_weight”: 2, “n_estimators”: 250, “scale_pos_weight”: 1.79, “subsample”: 0.75}.

The feature importance based on SHAP values is shown in Figure 2. The order of features reflects their weighted importance across the entire cohort (global explainability). The position on the y-axis is determined by the feature and on the x-axis by the Shapley value. The colour represents the value of the feature from low (blue = CC0 or yes) to high (red = not CC0 or no). The top-3 features included para-aortic lymph node dissection, UAP and pelvic lymph node dissection. Their longer tails compared to other features demonstrate their importance for specific in not all patients (local explainability).

When the features were screened using random forest, UAP was the top feature for CC0 prediction (Figure 3A). A correlation heatmap demonstrated the pairwise associations amongst the surgical procedures. The highest correlations were observed between large bowel resection and stoma formation ($r^{2}$ = 0.8), followed by bladder peritonectomy and pelvic peritonectomy ($r^{2}$ = 0.7). Satisfactory correlations were demonstrated between UAP and other surgical procedures. The UAP best correlated with bladder peritonectomy and diaphragmatic stripping ($r^{2}$ > 0.5) (Figure 3B).

The SHAP dependence plots reveal the impact of each feature on the prediction by plotting the value of the feature on the x-axis and the SHAP feature value on the y-axis. Certain surgical sub- procedures are clearly associated with higher likelihood of CC0 including stomach resection, UAP, diaphragmatic stripping (upper abdomen) (Figure 4A–C); small bowel resection, right hemicolectomy, stoma formation (bowel-related) (Figure 4D–F); all lymph node dissections ranging from para-aortic to groin dissections (Figure 5).

### 3.1. Model Comparison

When UAP was asked to predict solely CC0, The ROC curve showed that UAP could effectively distinguish cytoreductive outcome (AUC = 0.78, CI: 0.76–0.81). When UAP only was incorporated in a composite model comprising of engineered features, it substantially enhanced its predictive value (AUC = 0.80, CI: 0.76–0.84) (Figure 6).

### 3.2. Survival Data

The K-M analysis showed a difference between the CC0 and not CCO groups for both PFS and OS. The median PFS was 25 months for the CC0 group (95% CI 22–29) and 18 months (95% CI 17–19) for the not CC0 group (P < 0.05). The median OS was 58 months for the CC0 group (95% CI 55–62) and 33 months (95% CI 32–34) for the not CC0 group (P < 0.05).

In multivariate Cox analysis, performance of UAP was associated with poorer PFS (HR: 1.76; 95% CI: 1.14–2.70, P = 0.001) (Figure 7A). There was a trend towards poorer OS (HR: 1.06; 95% CI: 0.56–1.99, P = 0.86) (Figure 7B). Similar but very marginal worsening survival trend was observed for SCS on PFS (HR: 1.04, 95% CI: 0.95–1.13, P = 0.430 and OS (HR: 1.01, 95% CI: 0.89–1.15).

## 4. Discussion

Surgeons are significantly challenged by EOC heterogeneity. There is an increasing need for tools to better tailor treatment strategies by improving the predictions of the surgical outcomes. By scrutinizing a validated but exhaustive list of surgical sub-procedures outside the “box standard” surgery for ovarian cancer, we aligned with the recently published NICE guidelines on maximal cytoreductive surgery [19] and successfully quantified the complexity of surgery, as highlighted in our proposed classification algorithm (Figure 8). By categorising critical procedures, we highlighted the potential key role of upper abdominal peritonectomy (UAP), a complex and technically demanding surgical procedure. Using a large dataset of women with advanced EOC who underwent cytoreductive surgery, we developed and validated an ML algorithm, which demonstrates satisfactory predictive performance but more importantly, identifies UAP as the most important procedural indicator of CC0 in surgically cytoreduced EOC women. In contrast to the Aletti SCS, which supported an arbitrary allocation of a higher score for complex procedures [6], our devised ML model supported the feature selection and weighted importance of all surgical sub-procedures irrespective of the individual practice. Nevertheless, if solely used, it did not yield any survival benefit. We found that UAP best correlated with bladder peritonectomy and diaphragmatic stripping. That said, in selected patients, the procedrure should be offered not in isolation but as part of a “surgical package”.

The result is not surprising. The right upper quadrant is mostly affected by cancer metastases. Therefore, dissection of upper abdominal disease is critical at advanced EOC cytoreductive surgery. Fundamental anatomical knowledge and great expertise are required to appreciate the critical vascular landmarks prior to dissection [20]. Disease > 1 cm involving the upper abdomen above the greated omentum has been found in a recent study [21]. A comprehensive approach to surgical cytoreduction should incorporate upper abdominal resection [22]. We acknowledge that adequate exposure is critical to allow for complete resection. In our centre, initiation of the paradigm shift towards more complex multi-visceral surgery in the years 2016 and 2017, allows for a more thorough early intra-operative examination by mobilizing the liver and other organs and exposing the pouch of Morrison [12]. Diaphragmatic involvement is estimated in up to 40% of these cases [23]. Sugarbaker originally described various peritonectomy procedures, often warranted for maximal syrgical cytoreduction [24]. He best defined UAP as the resection of parietal and visceral peritoneum in the upper abdomen. On the right upper quadrant, that would involve stripping from the right subhepatic space and from the surface of the liver, in addition to the right hemidiaphragm; on the left upper quadrant, stripping over the left adrenal gland and pre-renal fat, lesser omentectomy in addition to the left diaphragm and spleen. Subphrenic peritonectomies on both sides allow for visualisation of the pancreas. Of those, lesser omentectomy with stripping of the omental bursa appears to be the most difficult due to the occurrence of vital structures. Radical peritonectomies with en-bloc resection of extensive widespread diaphragmatic peritoneal carcinomatosis have also been described [25]. Herein, we considered diaphragmatectomies as separate sub-procedures. Centralised surgical care is the best strategy to optimise oncologic outcoms with acceptable morbidity, even for those patients with high disease burden [26].

Overall, the study indicates that certain surgical procedures -and not the overall surgical load- are predictive of the likelihood for CC0. In addition to UAP, the top feature ranking was complemented by regional pelvic and para-aortic lymphadenectomies. Historically, nodal dissectios were associated with long-term survival [27]. Between 2014 and 2019, the results of the LION trial were not available [28]. Therefore, during surgical cytoreduction either on the upfront or delayed setting, the bilateral pelvic and para-aortic regions were systematically assessed, and consequently, systematic lymphadenectomy was rather routinely performed. Following publication of the LION trial results, routine lymphadenectomy is not warranted, as it does not confer a survival benefit unless there is evidence of macroscopically or radiologically enlarged lymph nodes. Disease distribution in the omental bursa, pancreatic surface, caudate lobe and portal trial are not absolute contraindications for debulking, unless there is deep infiltration of the porta hepatis or the celiac trunk [29].

The established benefit of upper abdominal cytoreduction in advanced EOC has been demonstrated even for optimal cytoreduction [30,31]. In our study, we failed to confer a survival benefit from the sole performance of UAP. At first glance, this looks odd. We explained why UAP should be offered as part of a “surgical package” to selected patients. It appears that any transient benefit is potentially outplayed by a high disease load in that cohort of patients. When discussing the potential benefits from UAP, the focus should drift from the hazard ratio to the shape of probability distribution, which is disease related. Although it can be helpful for the purposes of statistical hypothesis testing the benefit from the procedure, other measures such as median times to the study endpoint are important, particularly useful when the event of interest i.e., OS may eventually occur across the entire cohort. Then the risk for death is no longer an issue [32]. In our study, although UAP increased the hazard rate for PFS, the treatment effect was larger because >50% of the patients did not have a relapse at the time. Our CC0 are not inferior to those of other well established high-volume centers [31].

## 5. Strengths and Limitations

The study supported the current paradigm shift for organised centralisation of services moving away from the traditional patterns of cytoreductive surgery. Strength of the study was the study design that allowed to weight the importance of the individual procedures as outcome indicators. The cohort has been extensively scrutinised [12,13]. We applied XAI frameworks to explain the modelling “black box”, but also quantitative results not essentially included under the XAI umbrella, such as Cox regression [33]. We did not assess the morbidity of the surgical procedures, but it is assumed to vary as others have demonstrated the wide range in complications rates [34]. We are cautious about the generic application of our results in EOC modern care. Surgical experience and institutional capacity in the management of these patients may influence outcome rates and complications related to the incorporation of upper abdominal surgery [35]. Indeed, within our own practice, we observed variations in the surgical effort extended at complete resection. Nevertheless, the study was designed in such way not to reflect individual practice. Data from pre-operative imaging were not included in the study because the miliary or plaque-like morphology of the peritoneal disease makes it often undetectable by imaging [36]. Disease in the upper abdomen does not come without involvement of the lower regions [37]. To achieve complete clearance, we stress out the need for thorough exploration and visual inspection of the upper abdominal cavity early at surgery to resect all disease sites. Finally, if a robust surgical reference is to be generated in high output tertiary centres, a larger number of samples will be required.

## 6. Conclusions

We employed and explained an ML methodology for predicting the key surgical interventions required to achieve CC0. We identifed UAP as the most salient procedural indicator of CC0 in surgically cytoreduced EOC women. The upper abdominal quadrants should be thoroughly inspected to ensure that CC0 is achievable.

## Figures and Tables

**Figure 1 cancers-15-05386-f001:**
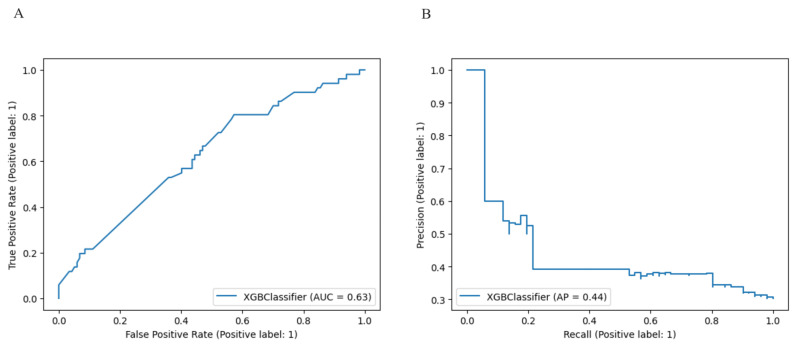
(**A**) Receiver Operator Characteristic (ROC) curve showing the diagnostic accuracy of all the surgical sub-procedures for the prediction of complete cytoreduction (AUC = 0.63) (**B**) Precision Recall curve and Average Precision performance value (AP = 0.44).

**Figure 2 cancers-15-05386-f002:**
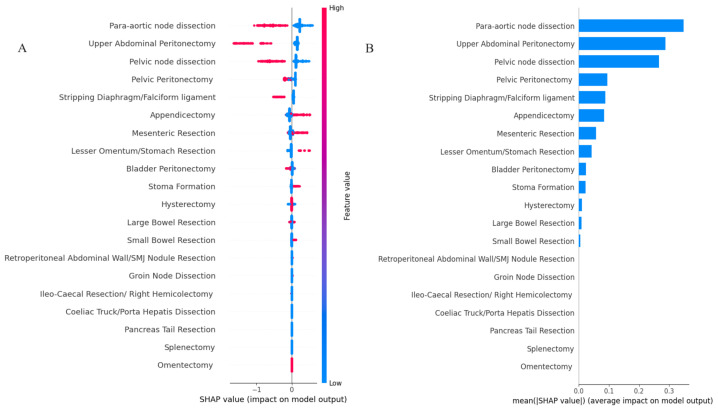
Model classification differences explained by the SHAP values. (**A**) Summary plot showing feature distribution plots based on the sum of SHAP value magnitudes over all samples. The color represents the feature value (Red not CC0 or no, Blue CC0 or yes resection) and the x-axis represents the impact score according to binary output (**B**) Standard bar plot of the mean absolute SHAP values for each feature showing the average impact on the global model output. SHAP, Shapley Additive explanations; CC, Complete Cytoreduction.

**Figure 3 cancers-15-05386-f003:**
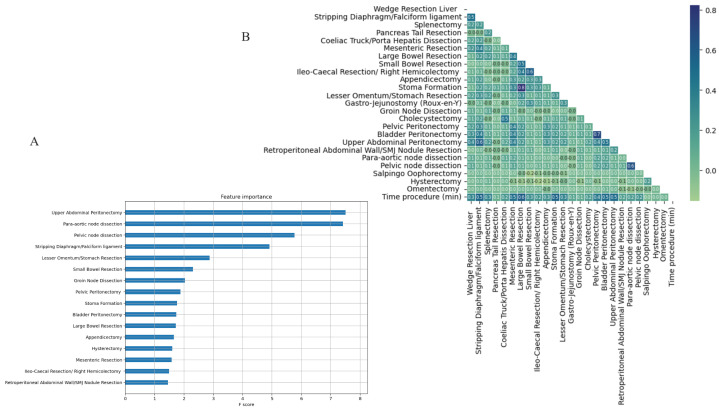
(**A**) Feature importance plot showing the relevance of each variable to the CC0 prediction when screened using random forest. (**B**) Correlation heatmap demonstrating the pairwise correlations amongst the surgical procedures. The Pearson correlation ($r^{2}$) was used. CC; complete cytoreduction.

**Figure 4 cancers-15-05386-f004:**
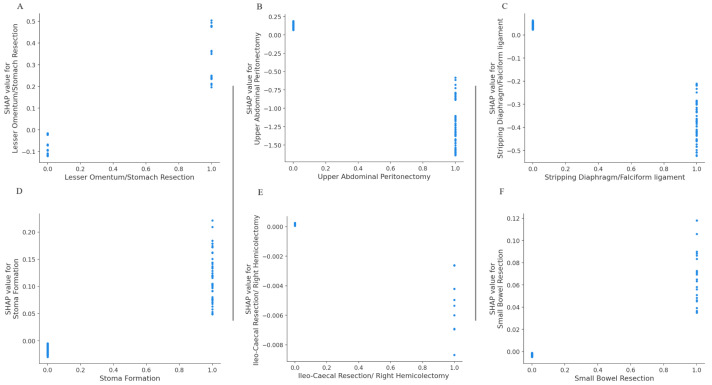
Dependence plots demonstrating clear inflection points for several surgical sub-procedures at cytoreduction (**A**–**C**) Upper abdomen, (**D**–**F**) Bowel resections.

**Figure 5 cancers-15-05386-f005:**
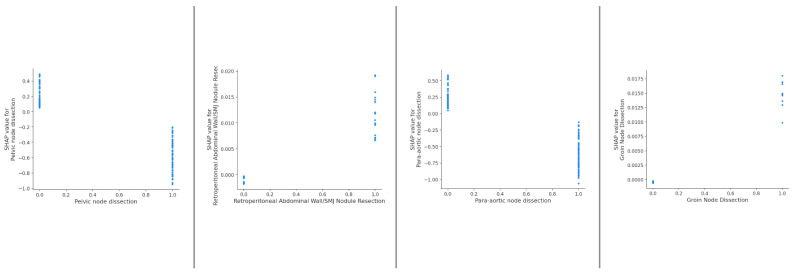
Dependence plots demonstrating clear inflection points for various regional lymph node dissections.

**Figure 6 cancers-15-05386-f006:**
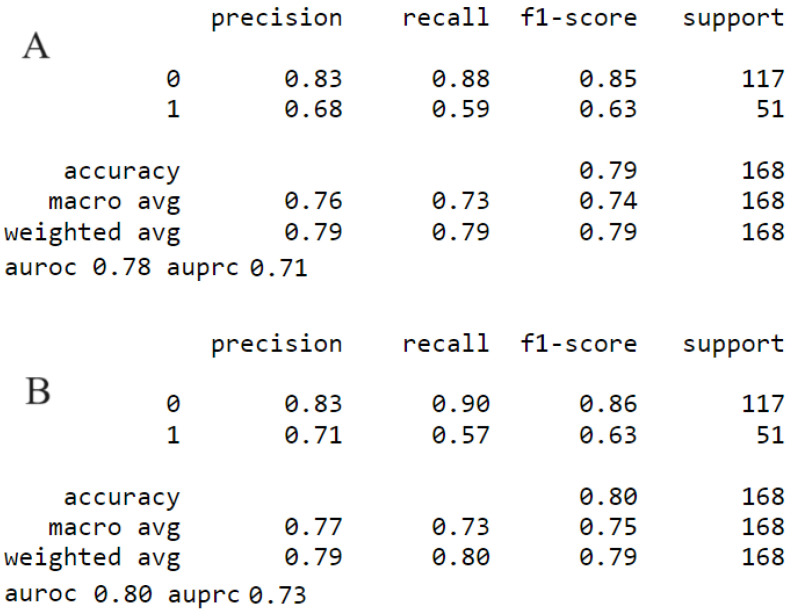
Performance metrics of devised models for the prediction of complete cytoreduction. (**A**) UAP. (**B**) Composite model comprised of UAP and commonly used engineered features.

**Figure 7 cancers-15-05386-f007:**
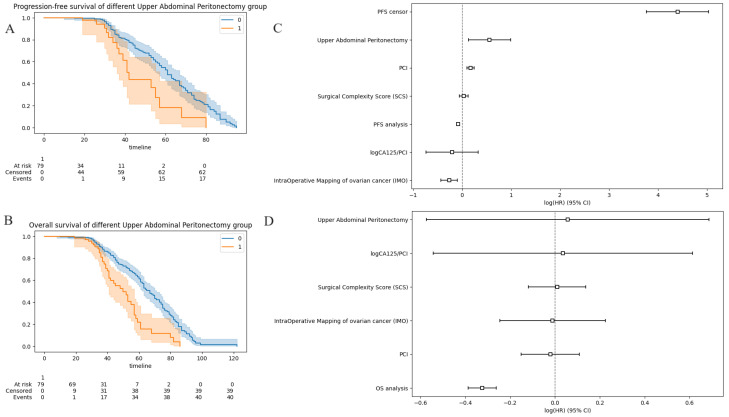
Cohort survival outcomes analyzed according to the occurrence of UAP (blue = UAP cohort; orange=non-UAP cohort) (**A**) progression-free-survival (**B**) overall-survival. Note the shape difference between the concave (UAP group) and the sinusoidal (non-UAP group) curves. Hazard ratio (HR) and 95% confidence interval (CI) for prospective log-linear associations (Cox regression) between (**C**) recurrence and non-recurrence (**D**) fatal and non-fatal outcomes including the UAP and commonly used engineered features. The shape of the curves rather than the hazard ratio can be used to quantify the benefit from the intervention. In contrast, a relatively small hazard ratio (concave curves) can yield large intervention effects reflected by longer median survival times for 50% of patients.

**Figure 8 cancers-15-05386-f008:**
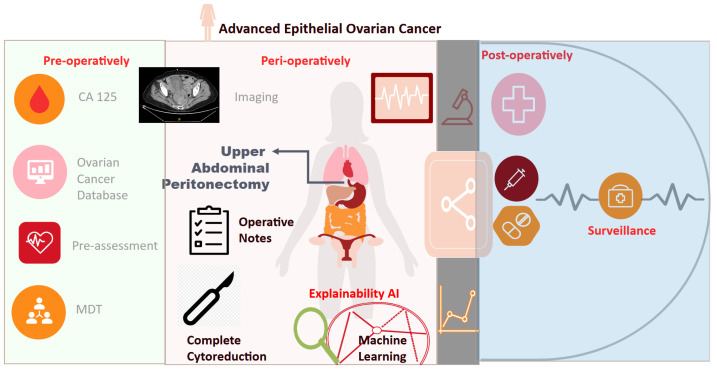
Study flowchart. The probability to achieve complete cytoreduction (CC0) can be well quantified by a ML-driven model inclusive all surgical sub-procedures. Upper abdominal peritonectomy is the most important predictive feature. A “surgical package” of maximal effort targeted cytoreduction including upper abdominal peritonectomy should be offered in selected patients. Thorough inspection of upper abdominal quadrants to ensure that CC0 is achievable reflects good clinical practice. ML: Machine Learning.

**Table 1 cancers-15-05386-t001:** Descriptive statistics of the performed surgical sub-procedures.

Variable		Overall (n = 560)	Training Set (n = 448)	Testing Set (n = 112)	P-Value	No UAP (n = 481)	UAP (n = 79)	P-Value
Wedge Resection Liver	0	537 (95.89)	428 (95.54)	109 (97.32)	0.558	475 (98.75)	62 (78.48)	<0.001
	1	23 (4.11)	20 (4.46)	3 (2.68)	0.558	6 (1.25)	17 (21.52)	<0.001
Stripping Diaphragm/Falciform ligament	0	484 (86.43)	385 (85.94)	99 (88.39)	0.6	455 (94.59)	29 (36.71)	<0.001
	1	76 (13.57)	63 (14.06)	13 (11.61)	0.6	26 (5.41)	50 (63.29)	<0.001
Splenectomy	0	543 (96.96)	435 (97.1)	108 (96.43)	0.951	474 (98.54)	69 (87.34)	<0.001
	1	17 (3.04)	13 (2.9)	4 (3.57)	0.951	7 (1.46)	10 (12.66)	<0.001
Pancreas Tail Resection	0	559 (99.82)	447 (99.78)	112 (100.0)	1.0	480 (99.79)	79 (100.0)	1
	1	1 (0.18)	1 (0.22)	0 (0.0)	1.0	1 (0.21)	0 (0.0)	1
Coeliac Truck/Porta Hepatis Dissection	0	554 (98.93)	443 (98.88)	111 (99.11)	1.0	479 (99.58)	75 (94.94)	0.002
	1	6 (1.07)	5 (1.12)	1 (0.89)	1.0	2 (0.42)	4 (5.06)	0.002
Mesenteric Resection	0	427 (76.25)	340 (75.89)	87 (77.68)	0.785	399 (82.95)	28 (35.44)	<0.001
	1	133 (23.75)	108 (24.11)	25 (22.32)	0.785	82 (17.05)	51 (64.56)	<0.001
Large Bowel Resection	0	496 (88.57)	399 (89.06)	97 (86.61)	0.572	440 (91.48)	56 (70.89)	<0.001
	1	64 (11.43)	49 (10.94)	15 (13.39)	0.572	41 (8.52)	23 (29.11)	<0.001
Small Bowel Resection	0	537 (95.89)	430 (95.98)	107 (95.54)	1.0	464 (96.47)	73 (92.41)	0.168
	1	23 (4.11)	18 (4.02)	5 (4.46)	1.0	17 (3.53)	6 (7.59)	0.168
Ileo-Caecal Resection/Right Hemicolectomy	0	539 (96.25)	432 (96.43)	107 (95.54)	0.868	465 (96.67)	74 (93.67)	0.326
	1	21 (3.75)	16 (3.57)	5 (4.46)	0.868	16 (3.33)	5 (6.33)	0.326
Appendicectomy	0	439 (78.39)	352 (78.57)	87 (77.68)	0.939	398 (82.74)	41 (51.9)	<0.001
	1	121 (21.61)	96 (21.43)	25 (22.32)	0.939	83 (17.26)	38 (48.1)	<0.001
Stoma Formation	0	509 (90.89)	407 (90.85)	102 (91.07)	1.0	449 (93.35)	60 (75.95)	<0.001
	1	51 (9.11)	41 (9.15)	10 (8.93)	1.0	32 (6.65)	19 (24.05)	<0.001
Lesser Omentum/Stomach Resection	0	534 (95.36)	427 (95.31)	107 (95.54)	1.0	468 (97.3)	66 (83.54)	<0.001
	1	26 (4.64)	21 (4.69)	5 (4.46)	1.0	13 (2.7)	13 (16.46)	<0.001
Gastro-Jejunostomy (Roux-en-Y)	0	558 (99.64)	447 (99.78)	111 (99.11)	0.859	480 (99.79)	78 (98.73)	0.658
	1	2 (0.36)	1 (0.22)	1 (0.89)	0.859	1 (0.21)	1 (1.27)	0.658
Groin Node Dissection	0	549 (98.04)	440 (98.21)	109 (97.32)	0.819	473 (98.34)	76 (96.2)	0.407
	1	11 (1.96)	8 (1.79)	3 (2.68)	0.819	8 (1.66)	3 (3.8)	0.407
Cholecystectomy	0	553 (98.75)	442 (98.66)	111 (99.11)	1.0	479 (99.58)	74 (93.67)	<0.001
	1	7 (1.25)	6 (1.34)	1 (0.89)	1.0	2 (0.42)	5 (6.33)	<0.001
Pelvic Peritonectomy	0	277 (49.46)	216 (48.21)	61 (54.46)	0.281	273 (56.76)	4 (5.06)	<0.001
	1	283 (50.54)	232 (51.79)	51 (45.54)	0.281	208 (43.24)	75 (94.94)	<0.001
Bladder Peritonectomy	0	358 (63.93)	284 (63.39)	74 (66.07)	0.676	351 (72.97)	7 (8.86)	<0.001
	1	202 (36.07)	164 (36.61)	38 (33.93)	0.676	130 (27.03)	72 (91.14)	<0.001
Upper Abdominal Peritonectomy	0	481 (85.89)	383 (85.49)	98 (87.5)	0.693	481 (100.0)	0 (0.0)	<0.001
	1	79 (14.11)	65 (14.51)	14 (12.5)	0.693	0 (0.0)	79 (100.0)	<0.001
Retroperitoneal Abdominal Wall/SMJ Nodule Resection	0	537 (95.89)	432 (96.43)	105 (93.75)	0.312	470 (97.71)	67 (84.81)	<0.001
	1	23 (4.11)	16 (3.57)	7 (6.25)	0.312	11 (2.29)	12 (15.19)	<0.001
Para-aortic node dissection	0	381 (68.04)	303 (67.63)	78 (69.64)	0.768	333 (69.23)	48 (60.76)	0.172
	1	179 (31.96)	145 (32.37)	34 (30.36)	0.768	148 (30.77)	31 (39.24)	0.172
Pelvic node dissection	0	414 (73.93)	335 (74.78)	79 (70.54)	0.427	363 (75.47)	51 (64.56)	0.056
	1	146 (26.07)	113 (25.22)	33 (29.46)	0.427	118 (24.53)	28 (35.44)	0.056
Salpingo Oophorectomy	0	6 (1.07)	3 (0.67)	3 (2.68)	0.182	6 (1.25)	0 (0.0)	0.683
	1	554 (98.93)	445 (99.33)	109 (97.32)	0.182	475 (98.75)	79 (100.0)	0.683
Hysterectomy	0	56 (10.0)	41 (9.15)	15 (13.39)	0.245	49 (10.19)	7 (8.86)	0.871
	1	504 (90.0)	407 (90.85)	97 (86.61)	0.245	432 (89.81)	72 (91.14)	0.871
Omentectomy	0	7 (1.25)	4 (0.89)	3 (2.68)	0.296	7 (1.46)	0 (0.0)	0.594
	1	553 (98.75)	444 (99.11)	109 (97.32)	0.296	474 (98.54)	79 (100.0)	0.594

## Data Availability

The data presented in this study are available on request from the corresponding author.

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
