# Peer review of "Exploring the Potential Role of Upper Abdominal Peritonectomy in Advanced Ovarian Cancer Cytoreductive Surgery Using Explainable Artificial Intelligence"

_cancers, 2023, doi:10.3390/cancers15225386_

Round 1

Reviewer 1 Report

Comments and Suggestions for Authors

Very interesting paper with practical conclusions. Practicing surgeons know about importance of upper abdominal surgery for CC0 success, however, the results of the study help to quantify the surgical efforts. Did authors perform laparoscopic estimation of operability prior to primary cytoreduction? If yes how many patients (%) had this procedure. What were the results of PCI index in the studied group? Could authors give some more details about the distribution of implants in the studied group, at least how many patients had upper abdominal localization and in what places precisely (table)?

Author Response

Comments addressed as per file

Reviewer 2 Report

Comments and Suggestions for Authors

The manuscript of the article is devoted to the actual problem of ovarian cancer. The manuscript uses many statistical methods for mathematical modeling and forecasting. The article has sufficient scientific value for publication in a journal.

I have minor comments on the manuscript; it may be possible to improve it.

- the presentation of the material is complex and dry; it is relatively difficult to read. In the introduction, I wanted more information about cytoreductive surgery.

- not everything is clear about the technical side of the application of artificial intelligence: I see statistical calculations, but the characteristics of the Shapley Additive explanations (SHAP) framework and its methods are not clearly described. Perhaps you should include information about computer equipment in Materials and Methods.

- keywords of the article are not specified in the abstract,

- figures 4 and 5 may need to be presented in a different form, as they have small details that are difficult to understand.

- it is better to revise the conclusion to be more specific since only the last sentence corresponds to this section.

Author Response

Comments addressed as per file attached

Reviewer 3 Report

Comments and Suggestions for Authors

I found the article as important and interesting. However, a number of minor changes are required.

Although the authors focus their work on UAP, there is no clear definition of UAP in their paper. Please define UAP

The authors performed the study to find the surgical sub-procedures related to CC-0 resection. However, the data are not presented clearly. Please look:

-          Figure 2. : lines 156 - 158 “The colour represents the value of the feature from low (blue = CC 0) to high (red = non-CC 0). The top-3 features included para-aortic lymph node dissection, UAP, and pelvic lymph node dissection” – these top procedures are highlighted in red. Does it mean, that the conduction of these procedures is related to non-CC0 (residual disease)?

-          However 161 – 162: “When the features were screened using random forest, UAP was the top feature for CC0 prediction (Figure 3A)”. Similarly, the results from the abstract indicate that performing UAP is associated with CC0. Lines 26-27: “Our study identifies UAP as the most important procedural predictor of CC0 in surgically cytoreduced advanced-stage EOC women. ”.

Therefore, the results should be rewritten and explained.

The discussion section is interesting, however, it does not correspond to the obtained results. I would suggest rewriting the discussion section and commenting on the results of the study.

There is no data describing the demographic characteristics of the patients. Although the authors indicate, that the data is available in their previous work, I will suggest incorporating it in this manuscript.

There is no caption for table 1 and table 2, and therefore, the tables are not clear. It is not clear to me, what the p-value refer to.

Figures 1 and 2 – lack of titles .

Author Response

Comments addressed as per file
